# N-Octadecane Encapsulated by Assembled BN/GO Aerogels for Highly Improved Thermal Conductivity and Energy Storage Capacity

**DOI:** 10.3390/nano13162317

**Published:** 2023-08-12

**Authors:** Siyue Hui, Rong Ji, Huanzhi Zhang, Chaowei Huang, Fen Xu, Lixian Sun, Yongpeng Xia, Xiangcheng Lin, Lei Ma, Hongliang Peng, Bin Li, Yazhen Wang, Erhu Yan, Pengru Huang

**Affiliations:** 1School of Material Science & Engineering, Guilin University of Electronic Technology, Guilin 541004, China; hsy1553676413@163.com (S.H.); ji.rong3@tfme.com (R.J.); 15077309361@163.com (C.H.); xufen@guet.edu.cn (F.X.); ypxia@guet.edu.cn (Y.X.); xclin2008@163.com (X.L.); malei@guet.edu.cn (L.M.); hlpeng2005@126.com (H.P.); li_bin@guet.edu.cn (B.L.); yeh@guet.edu.cn (E.Y.); pengruhuang@guet.edu.cn (P.H.); 2Guangxi Key Laboratory of Information Materials, Guangxi Collaborative Innovation Center of Structure and Property for New Energy and Materials, Guilin University of Electronic Technology, Guilin 541004, China

**Keywords:** phase change materials, graphene oxide, boron nitride, thermal conductivity, energy storage capacity

## Abstract

The rapid development of industry has emphasized the importance of phase change materials (PCMs) with a high latent-heat storage capacity and good thermal stability in promoting sustainable energy solutions. However, the inherent low thermal conductivity and poor thermal-cycling stability of PCMs limit their application. In this study, we constructed three-dimensional (3D) hybrid graphene aerogels (GBA) based on synergistic assembly and cross-linking between GO and modified hexagonal boron nitride (h-BN). Highly thermally conductive GBA was utilized as the supporting optimal matrix for encapsulating OD, and further implied that composite matrix n-octadecane (OD)/GBA composite PCMs were further prepared by encapsulating OD within the GBA structure. Due to the highly thermally conductive network of GBA, the latent heat of the composite PCMs improved to 208.3 J/g, with negligible changes after 100 thermal cycles. In addition, the thermal conductivity of the composite PCMs was significantly enhanced to 1.444 W/(m·k), increasing by 738% compared to OD. These results sufficiently confirmed that the novel GBA with a well-defined porous structure served as PCMs with excellent comprehensive performance offer great potential for thermal energy storage applications.

## 1. Introduction

Rapid global economic and industrial growth has resulted in a significant increase in fossil fuel consumption and serious environmental pollution, compelling researchers to develop renewable energy to improve energy efficiency [1]. Therefore, there is an urgent need to develop sustainable and renewable energy such as waste heat and solar energy. Thermal energy storage, which is critical for energy transformation and storage, has been extensively investigated in recent years.

Phase change materials (PCMs), which can be classified as organic, inorganic, and eutectic, are highly capable of storing and releasing thermal energy during the isothermal phase transition process, making them a prevalent and effective method for storing thermal energy in latent heat storage systems [2,3,4]. Due to their high price, complex synthesis process, and poor thermal reliability, eutectic PCMs are less frequently used. The disadvantages of inorganic PCMs, such as strong corrosion, large subcooling, and phase separation, also hinder their application. As a result, organic PCMs are preferred by researchers due to their high energy storage capacity, good physical and chemical stability, non-toxicity, low price, and small subcooling [5,6,7].

Alkanes, the most commonly used organic PCMs, have phase transition temperatures that can be easily regulated, as they are mainly related to the carbon number of the alkane chains [8,9]. Specifically, octadecane (OD) has attracted significant attention due to its suitable phase transition temperature, high thermal storage capacity, non-toxicity and safety, and good biocompatibility [10,11]. However, OD has disadvantages such as low thermal conductivity and susceptibility to melt leakage, resulting in limited practical applications [12,13,14,15]. Furthermore, the low thermal conductivity of PCMs hinders thermal diffusion in the material, affecting phase change interface motion and the ability to store/release heat [16]. An effective way to address this problem involves encapsulating organic PCMs in carriers, forming into promising shape-stable composite PCMs [17,18,19,20,21]. Among these supporting matrix materials, multi-porous materials are much more effective in encapsulating PCMs due to their low density, large specific surface area, and light weight.

Aerogels are typical low-density bulk materials with interconnected three-dimensional (3D) network skeletons, with ultra-high porosity and hierarchical pores at the micro-, meso-, and macroscopic scales [22,23]. Their skeletal and multilevel pore structures contribute to a variety of extraordinary physical properties such as ultra-low density, large porosity, high specific area, and excellent adsorption properties [24,25,26,27,28]. These properties make aerogels ideal supporting matrix materials for PCMs, providing them with a variety of functional applications in energy conversion, storage, and thermal management systems.

Graphene aerogel has inherent favorable properties such as light weight, high specific surface area, outstanding mechanical properties [29,30,31], excellent chemical and thermal stability, and remarkable light absorption capacity, offering significant possibilities for improving the shape stability and light absorption capacity of composite PCMs. Moreover, highly aligned graphene can form high-quality thermally conductive networks, further improving the thermal conductivity and stability of the composites [32,33]. Mehrali et al. [34] prepared shape-stable composite PCMs by impregnating paraffin wax into graphene nano-sheets, while Zhao et al. [35] prepared a 3D aerogel with graphene nanosheets to encapsulate PEG through vacuum impregnation. Wang et al. [36] prepared hybrid graphene aerogels by combining a MOF with graphene oxide to encapsulate lauric acid, and the prepared aerogel-based composite PCMs displayed high thermal/conductive conductivity. Yang et al. [32] fabricated GO/GNP aerogels to encapsulate paraffins, and the composite PCMs exhibited good thermal stability. Although these graphene-based composite PCMs have shown high thermal conductivity, the presence of numerous macropores in the aerogel will still decrease the loading content of PCMs. Furthermore, the inherent brittleness and structural collapse of graphene aerogels have seriously restricted the widespread application of PCMs. Therefore, improving the mechanical stability of the graphene aerogel is critical for composite PCMs to sustain stresses generated due to volumetric changes during the phase change process. Furthermore, GO often suffers from weak interactions with the nanoparticles and poses a significant challenge in terms of cross-linking. Therefore, the preparation of stable interconnected 3D aerogels or scaffolds through self-assembly remains difficult.

As a type of two-dimensional nanomaterial, boron nitride (BN) nanosheets are analogous to graphene, with good high thermal conductivity, excellent chemical and thermal stability, as well as extraordinary mechanical strength [37,38]. Therefore, by combining the advantages of BN and graphene, building stable interconnected 3D porous networks within graphene aerogels to fabricate BN/GO hybrid aerogels may serve as a suitable solution to the aforementioned issues. In addition, the interconnected network structure of the hybrid aerogel can increase thermal conductivity. The synergistic effect of BN and graphene can further enhance the PCM loading capacity in aerogels, and the chemical cross-linking of BN and GO can ensure the excellent mechanical performance of the BN/GO hybrid aerogels. As a result, the introduction of BN may be beneficial for fabricating thermally conductive, mechanically stable, and electrically insulating PCMs.

In this study, we constructed a 3D interconnected hybrid graphene/BN aerogel (GBA) to obtain the synergistic effect between GO and h-BN. Shape-stable GBA/OD composite PCMs with high thermal conductivity and high latent heat were fabricated by using hybrid aerogels as the supporting matrix to encapsulate OD. To fabricate the mechanically stable hybrid graphene/BN aerogel, BN was modified to improve its affinity and dispersion. Highly dispersed BN fillers provided more active sites for cross-linking with the GO nano-sheets, forming the 3D interconnected porous network of the BN/GO hybrid aerogels. As a result, the porosity and surface of the porous BN/GO hybrid aerogels were improved, achieving more PCM loading and high thermal conductivity. The dispersed BN nano-sheets in the porous network reduced the thermal resistance of the composite PCMs, which significantly improved the thermal conductivity of the aerogels. The results showed that the BN/GO hybrid aerogels effectively improved the shape stability and mechanical performance of the composite PCMs; and the crystallinity, thermal properties, and thermal conductivity of the GBA/OD composite PCMs were enhanced, indicating significant potential for thermal energy storage applications.

## 2. Materials and Methods

### 2.1. Materials

Graphite powder (99.6%, 200 mesh) was obtained from Xianfeng Nano (Hangzhou, China) Material Technology Co., Ltd. OD (99%) was obtained from Alfa Aesar (Shanghai, China) Chemicals Ltd. PVP (Mn = 1,300,000) and BN (1~2 µm) were obtained from Sigma-Aldrich (Shanghai, China) Trading Co., Ltd. KMnO_4_ (99.5%), NaNO_3_ (99.0%), H_2_SO_4_ (98%), H_2_O_2_ (30%), and EDTA (analytical pure) were obtained from Sigma-Aldrich (Shanghai, China) Trading Co., Ltd.

### 2.2. Modification of BN

An amount of 5 g of BN and 200 mL of NaOH solution with a concentration of 5 wt% was mixed in a 500 mL three-neck flask at 90 °C in a water bath with vigorous stirring for 2 h. Then, the mixture was filtered and washed until the pH value became neutral. After that, the product was placed in an oven at 80 °C for 24 h to obtain the modified h-BN.

### 2.3. Preparation of the GBA Aerogels

GO was first prepared using the modified Hummers method, where a certain amount of h-BN, 0.1 g of PVP, and deionized water were homogeneously mixed by sonication for 30 min. Then, 10 mg/mL of dispersed GO solution was uniformly mixed with the modified h-BN solution to form an emulsion, and 50 µm of EDTA was added to the emulsion as a reducing agent. The mixture self-assembled and was reduced through hydrothermal synthesis at 160 °C for 12 h. The samples were then washed and frozen in the refrigerator, and further freeze-dried at −50 °C for 2 days. 

Finally, the lyophilized aerogels were heated to 900 °C at a rate of 5 °C/min and maintained for 3 h to obtain the final carbonized GBA aerogel. Different GO and h-BN mass ratios were used, namely 5:1, 3:1, and 1:1, and a blank aerogel without h-BN was prepared for comparison. The obtained samples were denoted as GBA-1, GBA-2, GBA-3 and GA.

### 2.4. Preparation of the OD/GBA Composite PCMs

A certain amount of melted OD and dichloromethane as a binder solvent in moderation was vacuum absorbed into the prepared GBA in a beaker at 60 °C for 3 h, producing the OD/GBA composite PCMs, as shown in Figure 1. The mass ratio of GBA and OD was 1:9, and the obtained composite PCMs were denoted as OD/GBA-1, OD/GNA-2, OD/GBA-3, and OD/GA.

### 2.5. Characterization and Tests

Scanning electron microscopy (SEM, FEI, Quanta FEG 450) was used to observe the micromorphology of the GBA aerogels and OD/GBA composite PCMs at 20 kV acceleration voltage, and the fracture surfaces of the samples were sputtered with a thin layer of gold before observation. X-ray diffraction (XRD, Bruker-D8 Advance) and Fourier transform infrared spectroscopy (FT-IR, Nicolet 6700) were used to characterize the crystal structure and chemical composition. Porous structure and pore size distribution were characterized by using nitrogen isothermal adsorption/desorption analysis (AutosorbiQ2, Kantar, Florida, USA). A differential scanning calorimeter (DSC, TA Q250) was used to study the thermal properties of the samples, such as heat storage enthalpy, phase transition temperature, and thermal cycling stability, in a temperature range of −10 °C~110 °C at a heating rate of 10 °C min^−1^ under N2 atmosphere. A thermal conductivity analyzer (DZDR-S) with a transient planar heat source was employed to test the thermal conductivities of the samples.

## 3. Results and Discussion

### 3.1. Micro-Morphologies of GBA

Figure 2 presents the SEM images of GBA with different h-BN contents, indicating that all samples exhibited a hierarchically ordered and porous network structure due to π-π interactions and hydrogen bonds between the graphene and h-BN nanosheets, which ensured self-assembly through the hydrothermal synthesis and reduction process of GO. A large number of folds and porous structures in the aerogels facilitated the adsorption and encapsulation of the PCMs. Furthermore, the addition amount of h-BN showed a different effect on the micro-morphologies and porous structures of the aerogels. The graphene aerogels without h-BN displayed smooth lamellae and porous structures with partial folds on the graphene nanosheets, indicating a 3D porous structure. With the introduction of h-BN, tiny white particles were observed on the graphene nano sheets and the porous structure was not as large as the pure graphene aerogels. With a GO and h-BN mass ratio of 5:1, the white particles were unevenly distributed in the aerogels, and the porous structure was not interconnected due to the small addition amount of h-BN. In addition, the white particles in the GBA aerogels increased and thickened with an increase in h-BN. With a GO and h-BN mass ratio of 3:1, we observed that the white h-BN particles were homogeneously dispersed in the GBA aerogels, forming an interconnected porous structure, indicating a well-assembled structure between GO and h-BN. However, we observed that a large amount of h-BN accumulated and agglomerated in the GBA aerogels when the mass ratio of GO and h-BN was 1:1, and the porous structure of the GBA aerogels became disordered and clogged, restricting the formation of the interconnected thermal conductivity network. Therefore, the addition of a moderate amount of highly thermally conductive h-BN was beneficial for refining the 3D thermal conductive network of the GBA aerogels, indicating that the GBA aerogels with a GO and h-BN mass ratio of 3:1 possessed a well-constructed porous network, serving as an ideal supporting matrix for PCMs.

The SEM images further demonstrated the microstructures of the GA and GBA-2 aerogels. Pure GA clearly exhibited twisted sheet-like structure, indicating the cross-linked porous structure of graphene [39], and the observed morphology of GBA-2 was very similar to GA. Moreover, some spherical nanosheets with a uniform size and elliptical shape were observed on the graphene nanosheets, indicating that the h-BN particles were successfully introduced into the porous structure, and were stacked in an orderly manner and arranged in a well scaled shape. These results confirmed that the orderly stacking and interactions between h-BN and GO created an interconnected porous network structure with high thermal conductivity for the GBA aerogels.

### 3.2. Chemical Compositions of the GBA

The microstructures of GO, h-BN, and GBA with different proportions of h-BN were assessed by FT-IR spectroscopy, as shown in Figure 3. According to the FT-IR spectra of h-BN, the absorption peaks that appeared at 3439 cm^−1^ and 1624 cm^−1^ corresponded to the stretching and bending vibrations of the N-H groups, respectively, and the stretching vibration of B-N was located at 1409 cm^−1^ and 805 cm^−1^ [40,41]. In addition, the FT-IR spectra of GO showed characteristic peaks at 3437 cm^−1^ and 1391 cm^−1^, corresponding to O-H stretching vibrations, and the stretching vibration peaks of carbonyl and carboxyl C=O were observed at 1736 cm^−1^. The absorption peaks that appeared at 1624 cm^−1^ and 1127 cm^−1^ were associated with the O-H stretching vibrations. Compared to the FT-IR spectra of h-BN and GO, some of the characteristic peaks overlapped, such as the vibrations of B-N in h-BN and epoxy C-O in GO. According to the FT-IR curves of GBA containing different h-BN ratios, the characteristic peaks of both h-BN and GO appeared, indicating that h-BN was successfully introduced in the porous structure of the aerogels. These results further confirmed the successful synthesis of the GBA aerogels.

X-ray powder diffraction (XRD) was used to reveal the crystallization properties of GBA at room temperature, and the obtained XRD patterns of the GO, h-BN, and the GBA aerogels with different h-BN proportions are shown in Figure 3b. The characteristic diffraction peak of GO occurred at around 11.3°, corresponding to the typical sharp peak of (002) caused by the intercalation of the functional groups in the graphite layers. For h-BN, the diffraction peaks at 26.7°, 41.6°, 43.7°, 50.0°, and 55.1° corresponded to the crystal planes of (002), (100), (101), (102), and (004), respectively [42]. According to the XRD patterns of the GBA aerogels, the characteristic peak of GO disappeared because GO was reduced after self-assembly during the hydrothermal reaction and calcination, forming into the porous framework GA. In addition, a new diffraction peak appeared at 26.7°, which coincided with one of the diffraction peaks of h-BN, indicating the successful fabrication of GO and h-BN. The diffraction peaks of h-BN in GBA remained unchanged after hydrothermal reaction and calcination because h-BN possessed good chemical inertness and thermal stability. As a result, the intensity of the GBA diffraction peaks increased with an increasing h-BN content. In summary, h-BN was successfully cross-linked with GO, producing a well-defined porous framework for GBA with different h-BN proportions.

### 3.3. Adsorption Properties of GBA

The multi-porous structures of GBA, such as the specific surface area, total pore volume, and pore size distribution, were assessed according to the N_2_ adsorption–desorption method, as shown in Figure 4a. Pure GA and GBA both exhibited type IV isotherms, indicating that these aerogels had macro- and mesoporous structures. According to the BET model calculations, the specific surface areas of GA, GBA-1, GBA-2, and GBA-3 were 92.78, 101.24, 174.09, and 113.22 cm^2^/g, respectively. These results demonstrated that the addition of an appropriate amount of h-BN could increase the specific surface area of the aerogel and improve its adsorption performance. However, an excessive addition of h-BN would cause agglomeration, leading to a blockage of the 3D thermal conductivity network and a decrease in the specific surface area, thus reducing adsorption performance. As a result, GBA-2 exhibited the largest specific surface area among the samples due to its interconnected porous network. The pore size distribution of the samples was in the range of 3~5 nm, as calculated by the Barrett–Joyner–Halenda (BJH) model [43]. Therefore, the combination of an appropriate amount of h-BN with GO was beneficial for improving the porous structure and adsorption performance of the aerogels.

### 3.4. Micro-Morphologies of the OD/GBA

The SEM images were also used to investigate the morphology and microstructure of the composite PCMs with GBA as the supporting matrix, as shown in Figure 5. The folds and pores in the highly cross-linked network of the GBA aerogel surface decreased and became smooth, indicating that PCM-OD was effectively encapsulated into the porous structure [3]. In addition, OD completely occupied the aerogel pores, and the folds of the aerogel became significantly smoother as the OD content increased, with no obvious interface between OD and the aerogel, indicating their good compatibility. OD was completely bound by the 3D network structure of the aerogel, which improved the thermal conductivity and mechanical properties of the composite PCMs [3]. However, a high OD content reduced the performance of the composite PCMs, as indicated by the surface of the OD/GBA-3 composite PCMs, which became rough. This was due to the excessive amount of h-BN in GBA-3, which caused the 3D mesh structure to become blocked, resulting in a low absorption capacity that could not undergo encapsulation in the PCMs, and aerogel GBA-3 being unable to completely wrap the same mass of OD as the other aerogels. In addition, excess OD accumulated on the aerogel surface, resulting in some leakage during the melting process. Compared to the samples, the OD/GBA-2 composite PCMs exhibited an enormous surface structure with almost no accumulation of OD. These morphologies confirmed that the interconnected porous framework provided a good encapsulation effect for the obtained composite PCMs.

### 3.5. Microstructure Analysis of the OD/GBA Composite PCMs

Figure 6 presents the FT-IR spectra of the OD, OD/GA, and all OD/GBA composite PCMs. According to the pure OD spectrum, the broad bands at 2957 cm^−1^, 2932 cm^−1^, and 2853 cm^−1^ were attributed to the stretching vibrations of the methyl and methylene groups. The peaks at 1466 cm^−1^ and 1378 cm^−1^ were ascribed to the stretching vibrations of -CH_2_, and the in-plane vibration of -CH_2_ appeared at 720 cm^−1^. These characteristic peaks of OD were also observed in the FT-IR spectra of all OD/GBA composite PCMs, indicating that no chemical reaction occurred between OD and the GBA aerogels. Therefore, the GBA aerogels were encapsulated and maintained the OD microstructure.

An XRD analysis was used to characterize the crystal structures of the samples, as shown in Figure 6b. The characteristic diffraction peaks of OD appeared at the 2θ of 18.98°, 19.47°, 23.06°, 24.45°, 34.24°, and 39.29°, which corresponded to the (010), (011), (100), (111), and (110) crystal planes, respectively. The diffraction patterns of OD/GBA were similar to those of pure OD, which demonstrated that pristine OD and OD/GBA had similar crystalline structures and crystal cell types. Therefore, OD encapsulated within the cross-linking porous structure of the GBA aerogels retained its crystal structure well, ensuring the excellent thermal energy storage capacity of the composite PCMs for thermal energy storage applications.

### 3.6. Thermal Storage Properties of the OD/GBA Composite PCMs

The thermal properties of the synthesized OD/GBA composite PCMs were evaluated using a DSC analysis. The obtained curves are displayed in Figure 7 and the corresponding data are summarized in Table 1. In the obtained melting thermo-grams of OD/GBA, only one single, high-resolution melting peak was observed. Notably, the OD/GBA samples also showed good crystallization peaks during the cooling process. These results indicated that the synthesized OD/GBA composite PCMs possessed excellent phase change energy storage behavior, providing strong thermal energy storage properties for various applications. The melting temperature of the composite PCMs was between 30 and 32 °C, and the crystallization temperature was in the range of 19–22 °C. The melting temperature of the composite PCMs slightly decreased, and the sub-cooling degree also decreased with h-BN addition. This indicated that h-BN addition improved the thermal conductivity of the interconnected porous network and accelerated the thermal response rate of the interior OD within the GBA aerogels. These results further revealed that h-BN effectively cross-linked with GO and constructed the interconnected porous framework, which enhanced the phase transition behavior of the OD/GBA composite PCMs. However, when the h-BN content was 50 wt%, the melting temperature of the composite PCMs increased instead, because the excessive h-BN content led to accumulation and agglomeration, which disrupted the 3D thermal conductive network and restricted the phase transition rate of the interior OD. As shown in Table 1, the melting latent heat of the composite PCMs ranged from 203.6 to 218.8 J/g, and the crystallization latent heat ranged from 201.0 to 215.6 J/g. We observed that the latent heat of the composite PCMs slightly decreased with an increase in h-BN content due to the decline in OD weight percent, and no significant difference was observed compared to the theoretical value. This was because h-BN was strongly linked in the interconnected 3D network of GBA and could serve as a supporting skeleton to improve shape stability and thermal conductivity during the phase transition process of the PCMs. These results revealed that the GBA aerogel played a significant role in enhancing the thermal conductivity and shape stability of the composite PCMs, effectively improving their phase change behavior and energy storage capacity.

### 3.7. Thermal Cycling Stability of the OD/GBA Composite PCMs

The thermal reliability of the composite PCMs was determined using DSC thermal cycling tests. Figure 8a presents the DSC curves of the OD/GBA-2 composite PCMs after 100 melting and cooling cycles. We observed a negligible change in the phase transition temperature and endothermic/exothermic peaks, and the melting/crystal latent heat showed only slight changes before and after 100 thermal cycles. These results confirmed that OD was completely encapsulated in the GBA aerogel, and the composite PCMs exhibited excellent thermal cycling stability. Meanwhile, no changes in the characteristic peaks were observed according to the FT-IR spectra before and after 100 thermal cycles (Figure 8b), indicating that the chemical structure of the OD/GBA-2 composite PCMs was stable over 100 melting and cooling cycles. According to these results, we concluded that the prepared shape-stable composite PCMs possessed good thermal reliability and chemical stability.

### 3.8. Thermal Conductivity of the OD/GBA Composite PCMs

The thermal conductivity of PCMs will affect their thermal response rate and thermal energy modulation to the external environment. Figure 9a shows the tested thermal conductivity of the pure OD and OD/GBA composite PCMs, indicating that pure OD exhibited a very low thermal conductivity of only 0.1724 W/(m·K), making it necessary to improve the thermal response rate and expand the application range for thermal energy storage systems. The results showed that the OD/GA composite PCMs relied solely on the 3D porous graphene aerogel network to achieve a thermal conductivity of 0.8677 W/(m·K), which was 403% higher than pure OD. When h-BN was assembled with the GA aerogel, the thermal conductivities of the OD/GBA composite PCMs significantly improved, increasing by 564%, 738%, and 433% compared to pure OD. Notably, the thermal conductivity of the OD/GBA-2 composite PCMs improved to 1.444 W/(m·K). This result suggested that the thermal conductivity of the composite PCMs increased with the increasing of the mass ratio of h-BN in the aerogel, because h-BN was strongly linked with GO nano-sheets, forming efficient filler–filler overlaps that reduced interfacial thermal resistance. Therefore, the stable 3D interconnected network constructed by GO and h-BN significantly improved the thermal conductive properties of the composite PCMs. Nevertheless, when h-BN content increased to a certain extent, excessive h-BN accumulated and agglomerated, destroying the continuous thermal conductive network and resulting in a decrease in thermal conductivity. Therefore, at a GO and h-BN mass ratio of 3:1, the GBA aerogel exhibited a good continuous thermal conductive network, which enhanced the thermal conductivity of the composite PCMs and improved their thermal response rate and thermal energy storage capacity.

The thermal regulation performance and thermal management of OD, GBA-2 aerogel, and the OD/GBA-2 composite PCMs were tested to evaluate the thermal response rate of the composite PCMs using a high and low temperature oven. The obtained temperature–time curves, recorded as the ambient temperature was varied from −20 °C to 50 °C, are shown in Figure 9b. As seen from the temperature–time curves, the temperature of the samples increased linearly with increasing ambient temperature, and the temperature of the GBA-2 aerogel showed a faster rate than that of OD and the OD/GBA-2 composite PCMs in the whole heating process, implying that the fast thermal response rate of GBA-2 was due to the high thermal conductivity. Further, a temperature plateau appeared in the temperature range of 25–31 °C for pure OD and the OD/GBA-2 composite PCMs due to the phase change process of OD. Although the temperature increase rate of the OD/GBA-2 was a little higher than that of OD at the initial heating process, the temperature increase rate of the OD/GBA-2 was much faster than that of OD after heating for about 2500 s. This result indicates that the high thermal conductivity of GBA-2 enhanced the thermal response rate of the OD/GBA-2 composite PCMs. Moreover, the temperature decrease rate of OD/GBA-2 was faster than that of OD during the cooling process. The temperature plateau also presented at 26.5–29.1 °C, which is consistent with the DSC results during the crystallization process. This result further confirms that the high thermal conductivity of GBA-2 increases the thermal response rate of the OD/GBA composite PCMs.

## 4. Conclusions

To effectively improve the thermal conductivity and thermal energy storage capacity of PCMs, novel OD/GBA form-stable composite PCMs were successfully prepared in this study using OD as the phase change ingredient, and highly thermal conductive GBA was constructed through the synergistic assembly of GO and h-BN as the supporting matrix. The results showed that the constructed GBA aerogels possessed a continuous porous structure and a high specific surface area of 174.09 cm^2^/g due to the good cross-linking between GO and BN. Further, OD encapsulated in the GBA aerogels possessed excellent shape stability without leakage problems, which also well maintained its crystal structures. As a result, the melting latent heat of the OD/GBA composite PCMs was enhanced by 208.3 J/g, and the thermal conductivity significantly improved to 1.444 W/(m·K), which was 738% higher than OD. Therefore, the composite PCMs hold significant potential for applications in the fields of thermal energy storage.

## Figures and Tables

**Figure 1 nanomaterials-13-02317-f001:**
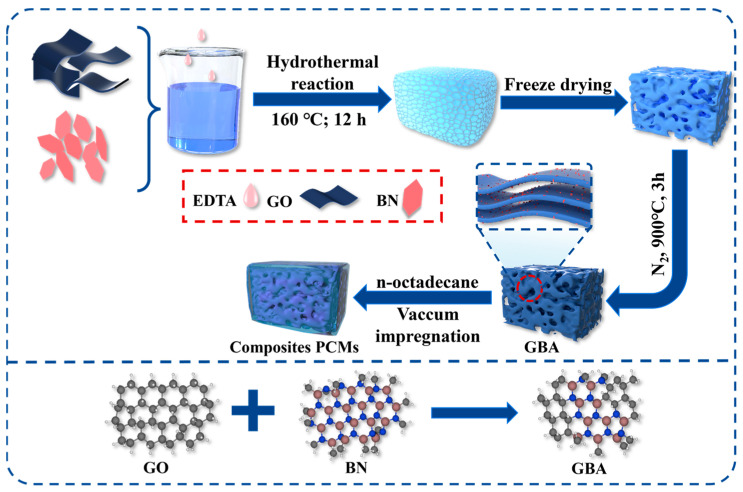
Schematic diagram of the OD/GBA composite PCMs.

**Figure 2 nanomaterials-13-02317-f002:**
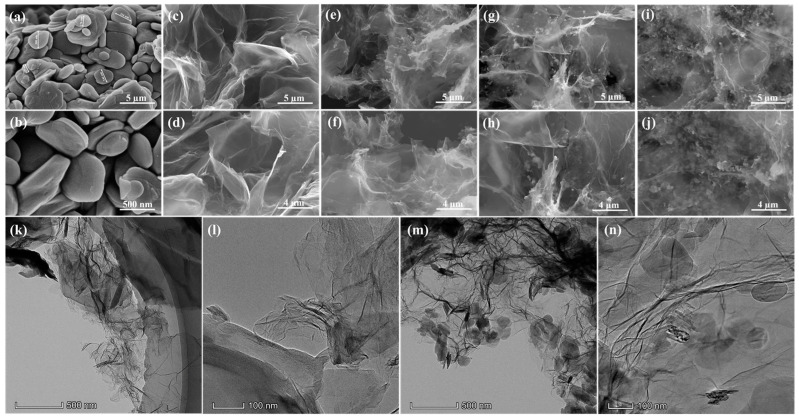
SEM images of GBA containing different h-BN content: (**a**,**b**) h-BN, (**c**,**d**) GA, (**e**,**f**) GBA-1, (**g**,**h**) GBA-2, (**i**,**j**) GBA-3; and TEM images of (**k**,**l**) GA and (**m**,**n**) GBA-2.

**Figure 3 nanomaterials-13-02317-f003:**
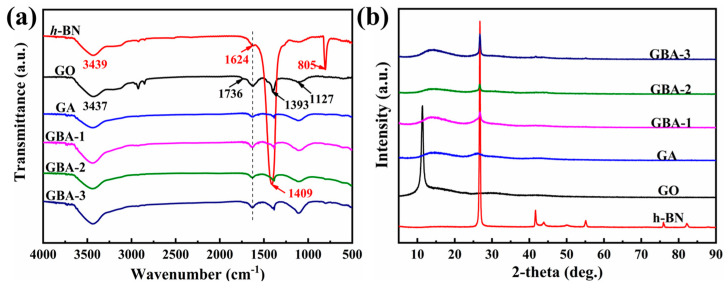
FT-IR spectra (**a**) and XRD patterns (**b**) of the GBA carbon aerogel with different GO contents.

**Figure 4 nanomaterials-13-02317-f004:**
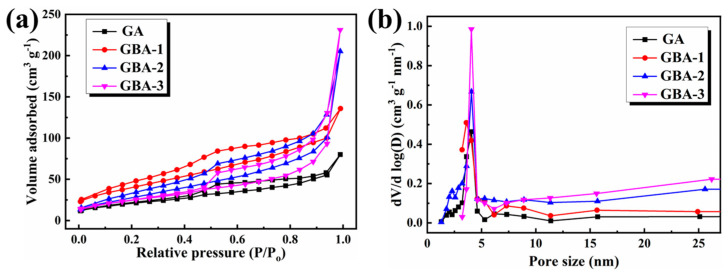
(**a**) Nitrogen adsorption–desorption isotherms and (**b**) pore size distribution.

**Figure 5 nanomaterials-13-02317-f005:**
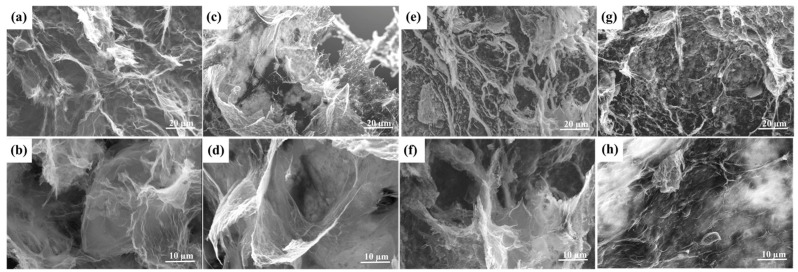
SEM images of the composite PCMs of (**a**,**b**) OD/GA, (**c**,**d**) OD/GBA-1, (**e**,**f**) OD/GBA-2, and (**g**,**h**) OD/GBA-3.

**Figure 6 nanomaterials-13-02317-f006:**
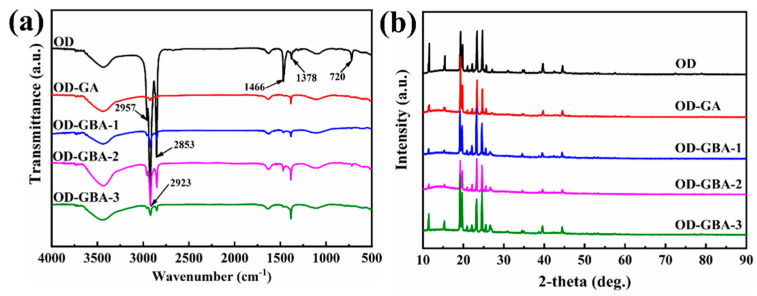
FT-IR spectra (**a**) and XRD patterns (**b**) of OD, OD/GA, and the OD/GBA composite PCMs.

**Figure 7 nanomaterials-13-02317-f007:**
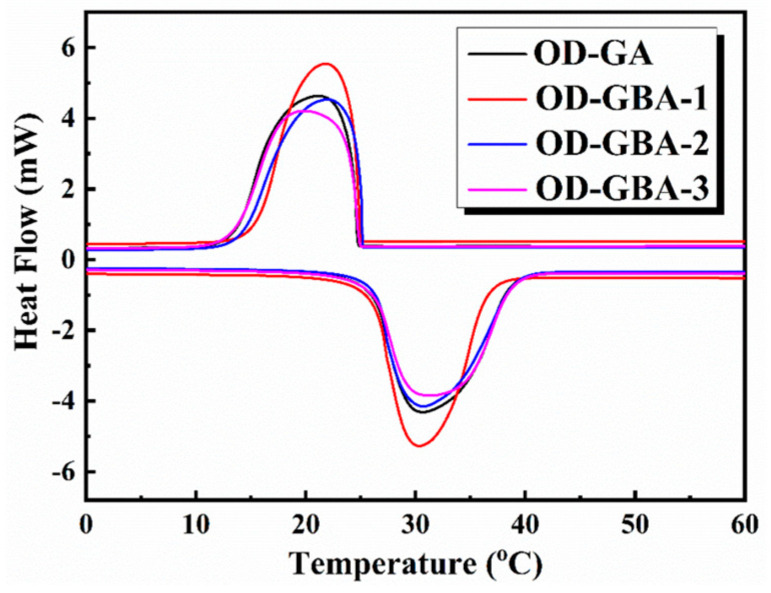
DSC thermograms of the obtained OD/GBA and OD-GA composite PCMs.

**Figure 8 nanomaterials-13-02317-f008:**
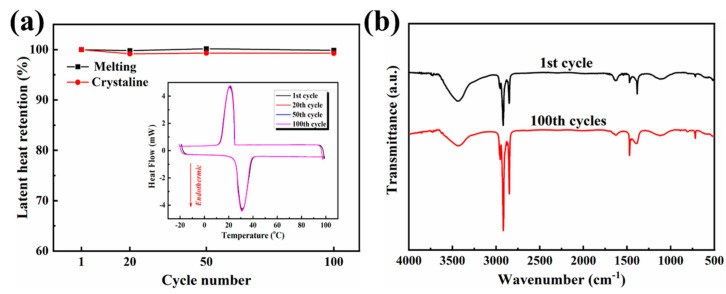
(**a**) The melting and crystal latent heat of the OD/GBA-2 experiencing 100 thermal cycles, and the inserted 100 times of DSC curves. (**b**) FT-IR spectra of the OD/GBA-2 before and after 100 thermal cycles.

**Figure 9 nanomaterials-13-02317-f009:**
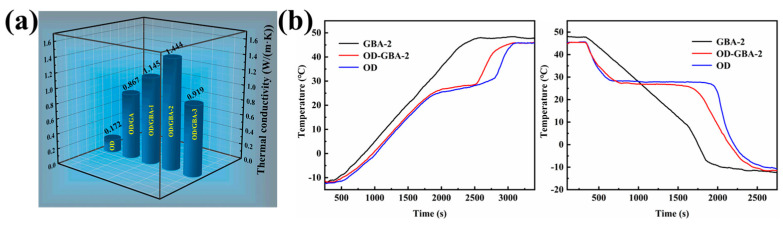
(**a**) Thermal conductivity of the OD and OD/GBA composite PCMs; (**b**) temperature–time curves of pure OD, GBA-2 aerogel, and the OD-GBA-2.

**Table 1 nanomaterials-13-02317-t001:** Phase change behavior of the synthesized OD/GBA composite PCMs.

Sample	Crystallization Process		Melting Process	
	*Tm* (°C)	Δ*Hm* (J/g)	*Tc* (°C)	Δ*Hc* (J/g)
OD	29.9	236.4	23.9	235.4
OD-GA	30.6	218.8	21.3	215.6
OD-GBA-1	30.4	214.4	21.8	212.2
OD-GBA-2	30.7	208.3	22.0	207.1
OD-GBA-3	31.4	203.6	19.8	201.0

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
