# Peer review of "N-Octadecane Encapsulated by Assembled BN/GO Aerogels for Highly Improved Thermal Conductivity and Energy Storage Capacity"

_nanomaterials, 2023, doi:10.3390/nano13162317_

Round 1

Reviewer 1 Report

Overall, the publication of this manuscript in the journal, Nanomaterials, can be positively evaluated; however, the following few points should be dealt with by the authors prior to the acceptance.

1. Comparative data of adsorption amount between pore space in GBA scaffold and actual OD is required. SEM image only knows qualitative data on this.

2. Under what conditions and by what mechanism does the crosslinking reaction take place between GO and n-BN? It is necessary to check the progress of the crosslinking reaction by appropriate analytic techniques.

3. Quantitative data on the thermal response rate due to thermal conductivity enhancement is required.

Native English edit with typo corrections is needed.

Author Response

We gratefully thank the editor and reviewer for making their constructive remarks and useful suggestions, which has significantly raised the quality of the manuscript and has enable us to improve the manuscript. Each suggested revision and comment, brought forward by the reviewers was accurately incorporated and considered. Below the comments of the reviews are response point by point and the revisions are indicated in the revised manuscript.

  1. Comparative data of adsorption amount between pore space in GBA scaffold and actual OD is required. SEM image only knows qualitative data on this.

Response: Thank you for your suggestion. In our experiment to prepare the OD/GBA composite PCMs, the adsorption amount of the OD has reached saturation point in the pore space of GBA scaffold. According to the weight calculation, the saturated adsorption amount is 93.2%, 91%, 89%, and 86.5%, for the sample OD/GA, OD-GBA-1, OD-GBA-2 and OD-GBA-3, respectively, which is consistent with the latent heat of the samples in DSC data as calculated based on the weight percent of OD.

  1. Under what conditions and by what mechanism does the crosslinking reaction take place between GO and n-BN? It is necessary to check the progress of the crosslinking reaction by appropriate analytic techniques.

Response: Thank you for your suggestion. As is well known, there are much functional groups of –OH, -COOH and epoxy groups on the surface of GO. And the modified BN also possesses –OH and –COOH groups, which strengthen the hydrogen bonds between GO and BN. Furthermore, the soft molecular chains of PVP were also introduced on the surface of modified BN, which increase the bonding energy and also ensure the crosslinking reaction between GO and h-BN. The hydrogen bonds between Go and BN were also added in Figure 1 in the revised manuscript. (Highlighted)

  1. Quantitative data on the thermal response rate due to thermal conductivity enhancement is required.

Response: Thank you for your suggestion. We have added the thermal regulation performance of the samples in Figure 9 to investigate the thermal response rate of the composite PCMs. and the corresponding descriptions were also added in the Figure 9 section.

 “Thermal regulation performance and thermal management of OD, GBA-2 aerogel and the OD/GBA-2 composite PCMs were tested to evaluate the thermal response rate of the composite PCMs using a high and low temperature oven, and the obtained temperature-time curves, recorded as the ambient temperature was varied from -20 °C to 50 °C, were shown in Figure 9(b). As seen from the temperature-time curves, the temperature of the samples increased linearly with increasing ambient temperature, and the temperature of GBA-2 aerogel showed the faster rate than that of OD and the OD/GBA-2 composite PCMs in the whole heating process, implying the fast thermal response rate of the GBA-2 due to the high thermal conductivity. And a temperature plateau appeared in the temperature range of 25-31 °C for pure OD and the OD/GBA-2 composite PCMs due to the phase change process of OD. Although the temperature increasing rate of the OD/GBA-2 is a little higher than that of OD at the initial heating process, the temperature increasing rate of the OD/GBA-2 is much faster than that of OD after heating for about 2500 s. This result indicates that the high thermal conductivity of the GBA-2 enhances the thermal response rate of the OD/GBA-2 composite PCMs. Moreover, the temperature decreasing rate of OD/GBA-2 is faster than that of OD during the cooling process. And the temperature plateau also presented at 26.5-29.1 °C, which is consistent with the DSC results during the crystallization process. This result further confirms the high thermal conductivity of GBA-2 increases the thermal response rate of the OD/GBA composite PCMs.”

In addition, the language of the whole manuscript has been polished by professional institutions, and some words have been corrected in the revised manuscript. (Highlighted)

Thank you very much again for your review and constructive suggestions.

Reviewer 2 Report

1. Authors need to show more of the SEM and TEM images of GBA and OD/GBA with only one image it is difficult to conclude the morphology. also please explain the morphology changes with different composite clearly.

2. The labeling in the figures is not clear, also the x- and y-axes labels

3. Please summarise the important findings in the conclusion

Author Response

We gratefully thank the editor and reviewer for making their constructive remarks and useful suggestions, which has significantly raised the quality of the manuscript and has enable us to improve the manuscript. Each suggested revision and comment, brought forward by the reviewers was accurately incorporated and considered. Below the comments of the reviews are response point by point and the revisions are indicated in the revised manuscript.

  1. Authors need to show more of the SEM and TEM images of GBA and OD/GBA with only one image it is difficult to conclude the morphology. also please explain the morphology changes with different composite clearly.

Response: Thank you very much for your suggestion. We have added some SEM images of GBA and OD/GBA composite PCMs in Figures 2 and Figure 5 in the revised manuscript. (Highlighted)

  1. The labeling in the figures is not clear, also the x- and y-axes labels

Response: Thank you for your suggestion. We have changed the x- and y-axes labels of the Figures in the revised manuscript. (Highlighted

  1. Please summarise the important findings in the conclusion

Response: Thank you for your suggestion. We have corrected the conclusion in the revised manuscript. (Highlighted)

“To effectively improve the thermal conductivity and thermal energy-storage capacity of PCMs, novel OD/GBA form-stable composite PCMs were successfully prepared in this study using OD as the phase change ingredient, and highly thermal conductive GBA was constructed through the synergistic assembly of GO and h-BN as the supporting matrix. Results showed that the constructed GBA aerogels possessed a continuous porous structure and a high specific surface area of 174.09 cm2/g due to the good cross-linking between GO and BN. And OD encapsulated in the GBA aerogels possessed an excellent shape-stability without leakage problems, which also well maintained its crystal structures. As a result, the melting latent heat of the OD/GBA composite PCMs was enhanced by 208.3 J/g, and the thermal conductivity significantly improved to 1.444 W/(m·K), which was 738% higher than OD. Therefore, the composite PCMs hold significant potential for applications in the fields of thermal energy storage.”

In addition, the language of the whole manuscript has been polished by professional institutions, and some words have been corrected in the revised manuscript. (Highlighted)

Thank you very much again for your review and constructive suggestions.

Reviewer 3 Report

The article contains a new approach to preparation of phase change materials which present a tool for saving the energy and lower tha actioon of the indusrtial development on the environment. The description of the preparation procedure and investigaton of physical properties of the materials produced are quite comprehensive. Therefore the article can be published. As a remark one shold note the graphene and BN samples used in the experiments have not been described. One should show the size and the number of layers in the samples.

The English language in the article should be improvved. For example the sentence on line 78 - "displayed high thermal/conductive conductivity" has to be proven. There are a set of such unproper sentences. 

Author Response

We gratefully thank the editor and reviewer for making their constructive remarks and useful suggestions, which has significantly raised the quality of the manuscript and has enable us to improve the manuscript. Each suggested revision and comment, brought forward by the reviewers was accurately incorporated and considered. Below the comments of the reviews are response point by point and the revisions are indicated in the revised manuscript.

  1. As a remark one shold note the graphene and BN samples used in the experiments have not been described. One should show the size and the number of layers in the samples.

Response: Thank you for your suggestion. We have added the size of BN and the SEM images of GO and BN in Figure 2 in the revised manuscript.

The h-BN dimensions have been labeled in Figure 2. The used raw material is 99.6% BN of 1~2 µm.

From the TEM and SEM images, it can be seen that our prepared GO in the laboratory has less layers of GO nanosheets, but due to the limited laboratory conditions cannot measure the specific number of the layers. It is about 8 layers by judgment based on the following literature.

Reference:

Vasanthi Shanmugam, S.; Palaniyandy, N.; Arumugam, K.; Venkatachalam, R. One Pot Green Synthesis of Few-Layer Graphene (FLG) by Simple Sonication of Graphite and Azardirachta Indica Resin in Water for High-Capacity and Excellent Cyclic Behavior of Rechargeable Lithium-Ion Battery. Diam Relat Mater 2023, 138, 110203, doi:10.1016/j.diamond.2023.110203.

In addition, the language of the whole manuscript has been polished by professional institutions, and some words have been corrected in the revised manuscript. (Highlighted)

Thank you very much again for your review and constructive suggestions.

Round 2

Reviewer 1 Report

(Comment #2) I can't still get the crosslinking reaction between GO and h-BN. Did the author mean physical crosslinking due to the hydrogen bonding between GO and h-BN?

Author Response

Dear reviewer,

     Thank you very much for reviewing our manuscript and proposing constructive suggestions, we have tried our best to explain the questions and have answered it, we hope this explanation can be accepted.

I can't still get the crosslinking reaction between GO and h-BN. Did the author mean physical crosslinking due to the hydrogen bonding between GO and h-BN?

Response: Thank you very much again for your constructive suggestion. The main physical cross-linking between GO and h-BN is the hydrogen bonding.

As it is well known, there are much functional groups of –OH, -COOH and epoxy groups on the surface of GO. And the modified BN also possesses –OH and –COOH groups, which strengthen the hydrogen bonds between GO and BN. Furthermore, the soft molecular chains of PVP were also introduced on the surface of modified BN, which increase the bonding energy and also ensure the crosslinking reaction between GO and h-BN. The hydrogen bonds between Go and BN were also added in Figure 1 in the revised manuscript.

In order to show clearly, we draw the crosslinking between GO and BN as shown in the following Figure.
